# De novo main-chain modeling for EM maps using MAINMAST

Genki Terashi [1] & Daisuke Kihara [1,2]

An increasing number of protein structures are determined by cryo-electron microscopy (cryo-EM) at near atomic resolution. However, tracing the main-chains and building full-atom models from EM maps of ~4–5 Å is still not trivial and remains a time-consuming task. Here, we introduce a fully automated de novo structure modeling method, MAINMAST, which builds three-dimensional models of a protein from a near-atomic resolution EM map. The method directly traces the protein's main-chain and identifies Cα positions as tree-graph structures in the EM map. MAINMAST performs significantly better than existing software in building global protein structure models on data sets of 40 simulated density maps at 5 Å resolution and 30 experimentally determined maps at 2.6–4.8 Å resolution. In another benchmark of building missing fragments in protein models for EM maps, MAINMAST builds fragments of 11–161 residues long with an average RMSD of 2.68 Å.

[1] Department of Biological Sciences, Purdue University, 249S. Martin Jischke Dr., West Lafayette, IN 47907, USA. [2] Department of Computer Science, Purdue University, 305N. University St., West Lafayette, IN 47907, USA. Correspondence and requests for materials should be addressed to D.K. (email: dkihara@purdue.edu)

An increasing number of protein structures have been determined by cryo-electron microscopy taking advantage of recent technological advancements which have enabled structure determinations at improved resolutions[1–3]. More EM maps were determined at near-atomic resolution in the past 2 years than in all other previous years combined: by the end of year 2015, only 114 EM maps had been released in the Electron Microscopy Data Bank (EMDB)[4] with resolution of 4 Å or better. In 2016 and 2017, 241 and 315 such high-resolution maps were released in EMDB, respectively, which totaled up to 731 at the end of 2017. The significant progress of the cyro-EM poses a pressing need for software for structural interpretation of EM maps, which include identifying a main-chain trace, building all-atom structure models, and model validation. Particularly, tools are needed for maps determined around 4 Å resolution, because maps near this resolution is often difficult to handle by software for X-ray crystallography[5,6] and also by conventional tools for EM maps that are for a lower resolution[7]. Considering that maps will be routinely determined in this resolution range for various proteins, tools are needed that can model structures de novo by detecting and tracing main-chain positions in an EM map without starting from fitting of existing template structures to the map. Compared with tools that aim to fit reference structures and refining a structure in EM maps[8–15], de novo modeling methods are still sparse in the field. Pathwalking is one such available de novo methods that constructs a protein Cα model from an EM map by connecting dense local map points using a Traveling Salesman Problem solver[16,17]. In Pathwalking, human intervention is needed for manual assignments of constraints and to determine the direction of protein sequence on the Cα model. Another de novo method is provided in the Rosetta protein modeling suite, which builds an initial model by assembling fragment structures taken from a protein structure database, followed by all-atom optimization to achieve better fit to an EM map[18,19].

Here we describe a de novo protein structure modeling method for EM maps of near atomic resolution. The method, named MAINMAST (MAINchin Model trAcing from Spanning Tree), has substantial advantages over existing methods: (i) MAIN-MAST directly constructs protein structure models from an EM density map without requiring reference structures; (ii) The procedure is fully automated and no manual setting is required; (iii) a pool of models are produced, from which a confidence level is computed that indicates accuracy of structure regions. We evaluated the MAINMAST's performance of constructing global protein structure models on two benchmark sets, a set of simulated density maps at a 5.0 Å resolution and a set of experimental EM maps. On the set of 40 simulated maps, MAINMAST produced high quality Cα models with an average root mean square deviation (RMSD) of 1.8 Å. On 30 experimental EM density maps of 2.6 to 4.8 Å resolution, on average MAINMAST produced models with 75.8% of Cα positions within 2.0 Å and 87.3% within 3.0 Å. These results were substantially more accurate than the two existing de novo methods, Pathwalking and Rosetta. MAINMAST was further tested in building missing fragments in proteins models for EM maps. MAINMAST showed comparable results with RosettaES building 44 fragments of 11 to 161 residues long at an average RMSD of 2.68 Å to the native conformation.

## Results

**Overview of the MAINMAST procedure**. MAINMAST builds protein main-chain structures from an EM map of around 4–5 Å or better by tracing local dense regions of the map. This method does not use existing structures including fragment structures because such an approach limits its application to globally or partially known structures and often has difficulty when constructing chains that contain uncommon conformations. When a map is determined at a resolution around 4–5 Å or better, the majority of the main-chain can be recognized in a map as dense regions. Figure 1 illustrates the overview of MAINMAST's procedure. The procedure starts by identifying local dense points in a given EM map, which are likely to correspond to the main-chain and sidechains of a protein. Then, these points are connected into a tree structure in a way that the total distance of connected points is minimized (i.e., the minimum spanning tree; MST). It was found that the main-chain of the protein is well covered by the MST because the number of points is large enough so that neighboring points are found in a short proximity to one another. The MST is constructed in two steps, first by constructing local MSTs for local regions around dense points to capture local topology of the chain, followed by construction of the MST that connects all the points using the local MSTs as constraints. Once the MST is constructed, it undergoes extensive conformational modification using an efficient search method, a tabu-search, to generate a pool of alternative trees. Trees are generated by changing parameters for defining density points and branches in a tree. The generated trees are then finally ranked with a score called the threading score, which evaluates the agreement to the density of each amino acid in the protein sequence. The top 500 Models selected by the threading score are subject to full-atom reconstruction and refinement using PULCHRA[20]. Finally, the full-atom models are refined using molecular dynamics flexible fitting (MDFF)[21], a molecular dynamics-based method, and selected according to the scoring function implemented in MDFF. Refer to Methods for more details.

**Models built for simulated EM maps**. First we evaluated the performance of MAINMAST on a data set of 40 simulated density maps, which were originally used by the paper of Pathwalking[16,17]. The maps were generated at a resolution of 5.0 Å with a grid spacing of 1.0 Å/voxel using the e2pdb2mrc.py program in the EMAN2 package[22]. The 40 proteins include 5, 20, 14, and 1 structures from the α, β, α/β, and the few secondary structure classes, respectively, according to the CATH protein structure classification database[23].

For each of the 40 maps, MAINMAST built 2688 Cα models with different parameters settings, which were ranked by the threading score that examines the agreement of the protein sequence to a Cα model. MAINMAST constructed accurate Cα models with an average RMSD of 1.79 Å to the native structures for all the 40 maps (Supplementary Table 1). Comparing the top scoring models by the threading score and overall the best (i.e., the smallest RMSD) models among 2688 models generated, it is found that the threading score was very successful in selecting models that are close to the best choice. On average, the RMSD between the best model in the model pool and the selected top-scoring model was only 0.39 Å (Supplementary Table 1).

Comparison against structure models constructed by Pathwalking[16] ver. 2016 is shown in Fig. 2a, b. For the Pathwalking algorithm, data are taken from the 2016 publication[16]. The figures indicate that MAINMAST built significantly better models than Pathwalking. For this performance comparison, the CLICK method[24] was used for evaluating models because it was used in the Pathwalking papers[16,17]. CLICK identifies similar local structures between a model and its native structure that have consistent inter-residue distance, the secondary structures, and the residue exposures, and computes RMSD (Fig. 2a) for the common local structures. It also reports the fraction of residues that are within 3.5 Å when the two structures are overlapped (Fig. 2b). Figure 2a shows that most models by

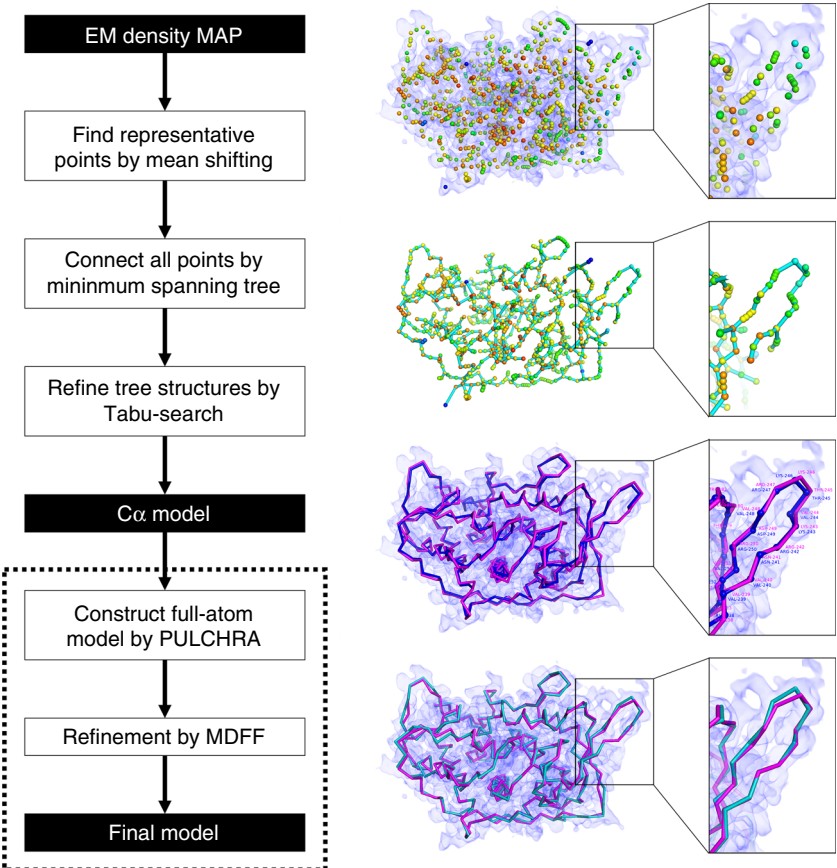

**Fig. 1** Flowchart of MAINMAST. Steps of the MAINMAST algorithm is illustrated with a modeling example for an EM density map of structural protein 5 of cytoplasmic polyhedrosis virus solved at a 2.9 Å resolution (EMD-6374). First, points with high local density are identified with the mean shift algorithm. The color scale of the points indicate density, blue to orange for low to high density. Identified local dense points are connected by minimum spanning tree (MST) (cyan). Using tabu-search, the initial MST is refined and a few thousands of alternative MSTs are generated. For each MST, the amino acid sequence of the query protein is mapped on the longest path in the tree by matching the volume of amino acids to the density of the local dense points (threading). Cα models from each MST are ranked with the density–volume matching (threading) score. In the third panel on the right, the blue chain represents a Cα model and the structure in magenta is the native structure. Selected Cα models are refined with a sequential application of PULCHRA and MDFF to obtain final full-atom models (turquoise)

MAINMAST have lower local RMSDs than Pathwalking models. Consistent with the results in Fig. 2a, Fig. 2b shows that MAINMAST models have a larger overlap to native than Pathwalking models for most of the cases. Supplementary Fig. 1 provides comparison with two versions of Pathwalking, ver. 2012 and ver. 2016.

The models by MAINMAST were further compared with those built with Rosetta (Fig. 2c, d). Using the Rosetta package, fragment structures were first assigned to a map by the Rosetta fragment assembly protocol, denovo_density, followed by RosettaCM[25], which builds a full residue models by filling gaps between fragments and optimizing the whole structure. The details of the modeling steps and commands used in the Rosetta package are provided in the Supplementary Note 1. The global Cα RMSD of the MAINMAST and Rosetta models are shown in Fig. 2c, while Fig. 2d presents the distribution of the fraction of residues in the native structure that were modeled within 2.0 Å to the correct positions (model coverage). In the two figures, top-scoring models (solid circles) as well as the best model among the top 10 scoring models (open circles) were compared. Among the 40 maps tested, MAINMAST achieved a smaller Cα RMSD than Rosetta on 26 maps, while the Rosetta's model was better for the rest of 14 maps when the top 1 models were considered (Fig. 2c). RMSD values of the models by MAINMAST distributed in a small range between 0.95 and 2.92 Å. On the other hand, Rosetta

generated high-quality models of less than a 1.0 Å RMSD for the 14 models that were better than MAINMAST; however, it also generated significantly high RMSD models. There were 21 Rosetta models which had an RMSD over 10.0 Å. Where the coverage is concerned, all MAINMAST models had a coverage over 0.9, while that of Rosetta models had a wider variation, from 0.28 to 1.0 in the coverage (for top 1 models, solid circles) (Fig. 2d). Interestingly, however, when MAINMAST and Rosetta models were compared for each map, Rosetta models had a larger overlap for 22 cases, when top 1 models were considered. To understand the nature of the Rosetta models, we further analyzed them by evaluating accuracy assigned fragments and full residue models in Supplementary Fig. 2. It turned out that the Rosetta protocol assigns fragment structures to an EM map accurately most of the time, but often could not assign fragments for the entire protein structure in the map, which led to inaccurate modeling in the subsequent step of filling gaps between assigned fragments.

In the four bottom panels in Fig. 2, we further examined the performance of the MAINMAST procedure. Figure 2e, f shows how well the threading score ranked Cα models. Threading scores and RMSDs of models for an EM map have a negative correlation since the higher threading scores are better. Figure 2e is a histogram of correlation coefficient between threading scores and RMSDs of models for the 40 EM maps. A negative correlation

was observed for all of the maps, including 29 maps that have a correlation higher than −0.5. Figure 2f shows such an example, threading scores and RMSDs of models for ferripyochelin binding protein (PDB: 1V3W) (correlation coefficient: -0.767).

The last two panels show the extent to which Cα models by MAINMAST can be further refined. Model refinement was performed in two steps. From a Cα model, a full atom model is

constructed using PULCHRA[20], which is then refined with MDFF[21]. Figure 2g demonstrates that the PULCHRA-MDFF refinement procedure improved Cα positions (Fig. 2g) over models for 28 out of 40 cases (70.0%). A larger improvement was observed for models that have a relatively high quality, those with Cα RMSD less than 2.0 Å. On an average of the 40 cases, the refinement procedure improved Cα RMSD from 1.79 to 1.66 Å.

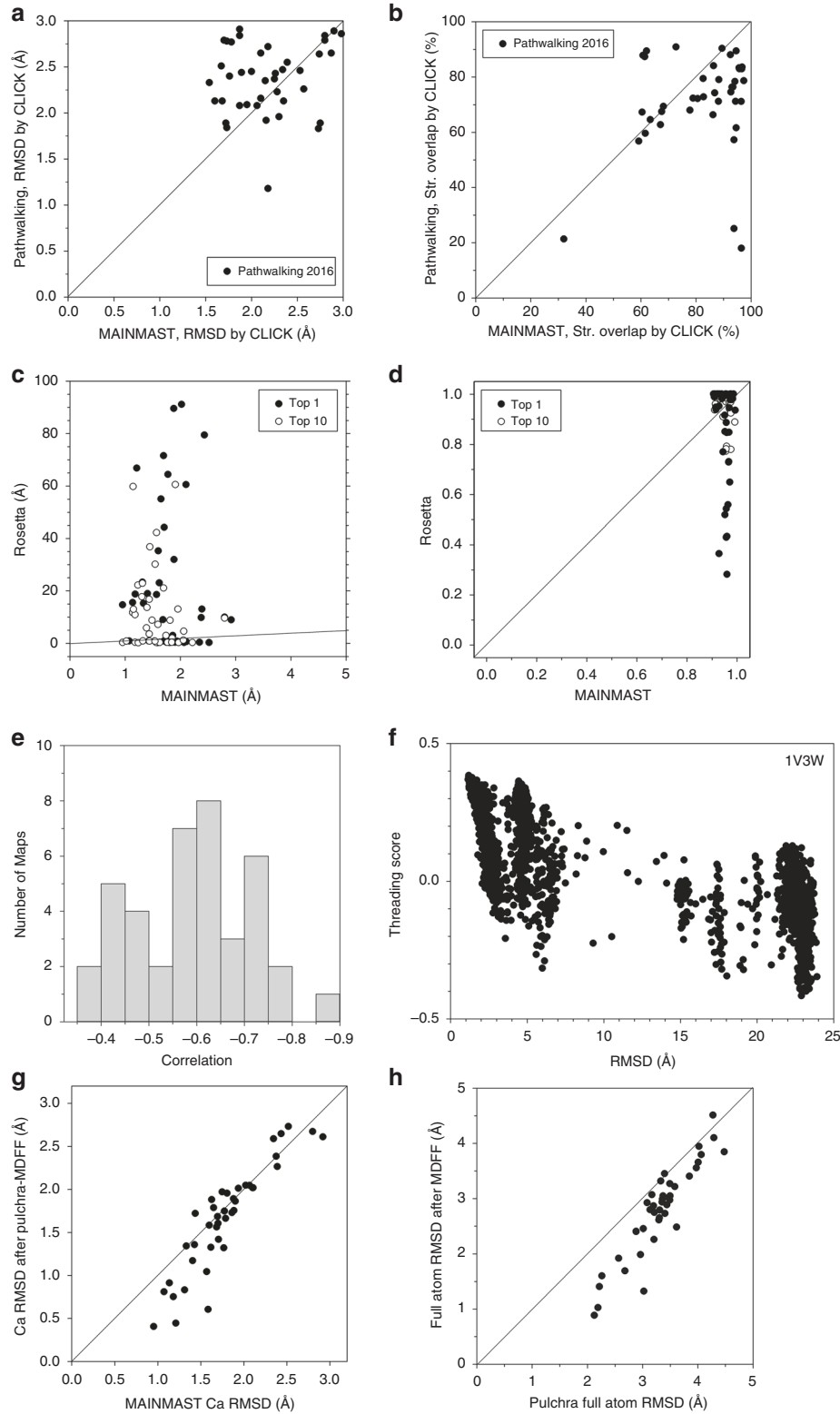

Comparing before and after MDFF (Fig. 2h), structures were improved for all but two cases. Note that PULCHRA and MDFF do not need human intervention such as manual parameter setting. Thus, the entire modeling procedure from the Cα model tracing by MAINMAST to the refinement is automated.

**Models built from experimental maps**. Next, we tested the methods on the second benchmark set of 30 experimental EM density maps obtained from EMDB (Supplementary Table 2). The resolutions of the maps ranged from 2.6 to 4.8 Å. The protein structure models deposited in PDB were used as the native reference structures. For each density map, a single subunit was manually segmented from a whole density map using UCSF Chimera's "zone tool" with the deposited structure[26]. For performing the Rosetta de novo modeling protocol, a single subunit was segmented out from the whole map at a distance of 4.0 Å as described in the Rosetta paper[19]. With MAINMAST, 500 full-atom models were built for each EM map, which were ranked by the scoring function used by MDFF (See Methods). With Rosetta, two parameter settings were used; the default setting[19] and another setting with a relaxed value, 0.8, for the consensus_frac parameter, which was used in the fragment mapping phase. With 0.8 of this parameter, fragment conformation of a position in an EM map is kept if 80% of the assigned fragments have consistent structures. The default is 1.0.

Figure 3a shows RMSD of models built by the MAINMAST procedure (Fig. 1) in comparison with models by Rosetta. For Rosetta, results using 0.8 for consensus_frac are shown because it gave better results than the default setting (Supplementary Fig. 3). The detailed results are provided in Supplementary Table 3. The average RMSD value of the MAINMAST models was 18.3 Å and 13 out of 30 top scoring (top1) models had an RMSD of less than 10.0 Å. On the other hand, Rosetta failed to model structures for two maps (points above the panel) and the average RMSD of the remaining 28 models was 27.0 Å. Rosetta often made models with over 50 Å RMSD and there were only six top 1 models that have an RMSD of less than 10.0 Å. Figure 3b presents sequence-alignment free quality measures, i.e., the fraction of residues in the native structure that are close (within 3.0 Å) to any residues in the model (coverage) and vice versa (precision). All MAINMAST models have both measures over 0.6 (average coverage, 0.88; and average precision: 0.89) while the fraction of Rosetta models varies from 0.17 to 0.99 with an average of 0.69. Thus, overall MAINMAST was more successful than Rosetta.

The latter three panels examine the performance of MAINMAST in details. Comparison between top 1 and top 10 models (Fig. 3c) shows that the model selection was successful when the structure pool contained a high quality model with an RMSD of less than 10.0 Å while top 10 choices had better models by often over 10 Å than the top 1 model when the quality of top 1 model was not high. Figure 3d summarizes the results of structure refinement with PULCHRA followed by MDFF. For the majority of the cases the refinement improved the structures including

cases with drastic improvements of over 20 Å RMSD. The largest improvement was achieved for the top 1 model of EMD-6478, where the model was improved from 40.9 to 3.7 Å RMSD. The large improvements occurred when the MDFF score selected better (lower RMSD) structures than the selection made by the threading score before the refinement. The model quality by MAINMAST showed a weak correlation to the map resolution (Fig. 3e).

Figure 4 illustrates models built by MAINMAST. The first model is for EMD-6555, an EM map of the porcine circovirus capsid protein determined at a 2.9 Å resolution. This protein has a β sheet with eight strands, which is in general difficult to trace by a software, as also evidenced by a 31.6 (30.4) Å RMSD model generated by Rosetta (the result with a 0.8 consensus setting is shown in parenthesis). Despite of the difficulty, the top 1 model by MAINMAST correctly traced the main-chain, yielding a 2.4 Å model. The next one (Fig. 4b) is another successful example of MAINMAST. This is a structure of the magnesium channel CorA, whose density map was determined at a 3.8 Å resolution (EMD-6551). In the figure, we showed two models from MAINMAST, the final top 1 model (turquoise) as well as the Cα model (blue) for which the full-atom building and refinement was applied to yield the final model. RMSD values of the models to the native were 3.8 and 4.4 Å, for the final and the Cα model, respectively. As shown in the figure, the refinement step substantially improved structures at helical regions of the protein, reproducing helical pitches close to the native. Rosetta models had an RMSD of 20.4 (12.3) Å. The next example, Fig. 4c is a model by MAINMAST for eL6 protein from yeast 60 S ribosomal subunit (EMD-6478, 2.9 Å resolution). In this case, the top-scoring Cα model by the threading score (blue) aligned the amino acid sequence in the opposite direction, resulting in an RMSD of 40.9 Å. Coverage and precision of this model are 0.73 and 0.74, respectively, which are not as low as one may think from the RMSD value because the traced path is almost on the main-chain. There was also a shift in the sequence mapping, which also contributed to the large RMSD. However, the model selection using the MDFF score after the full-atom building and refinement managed to select the near-native model, which has an RMSD of 2.6 Å. Rosetta models had an RMSD of 21.3 (10.6) Å for this protein. Figure 4d is a case that modeling by MAINMAST failed at a local region due to a low local map resolution (Measles virus nucleocapsid protein, EMD-2867, resolution 4.3 Å). The modeling was successful in the N-terminal to the middle part of the protein (the left side of the figure), but the main-chain from residues 32–34 and 93–95 (the zoomed region in the figure) were incorrectly traced. It turned out that the local resolution for these residue positions are significantly lower, 5.6–6.6 Å according to the ResMap program[27], which likely caused this problem.

**Confidence level of models**. As MAINMAST generates a large number of structure candidates, consensus among top scoring models are more reliable than other parts. Figure 5a, b shows the

**Fig. 2** Modeling results of the 40 simulated maps by MAINMAST in comparison with Pathwalking and Rosetta. **a** local RMSD and **b** structure overlap of the models by MAINMAST compared with Pathwalking models computed with the CLICK server. For the Pathwalking algorithm, data are taken from the publication in 2016. For the MAINMAST results, the model with the best threading score among the generated 2688 models were used. Structure overlap by CLICK in panel **b** is defined as the percentage of residues in a structure placed within 3.5 Å to residues in the other superimposed structure. **c**, **d** show comparison of the models by MAINMAST and Rosetta in terms of **c** the global RMSD and **d** the coverage, which is defined as the fraction of residues in a model that have some residues in the model within 2.0 Å. Solid /open circles, the highest scoring models/the best models among generated models were used for MAINMAST and Rosetta, respectively. Lines show $y = x$. **e** A histogram of correlation coefficients between the threading scores and RMSD of 2688 models generated by MAINMAST for the 40 EM maps. The correlation coefficient values are negative because the threading score is a high positive value for a near native model with a small RMSD. **f** correlation between the threading scores and RMSD values of models generated for 1V3W. The correlation coefficient is -0.767. **g** Comparison of Cα RMSDs of models before and after the full atom reconstruction and refinement using Pulchra and MDFF (g-scale used was 0.5) for the 40 maps. **h** Comparison of full-atom RMSDs of models before and after structure refinement by MDFF

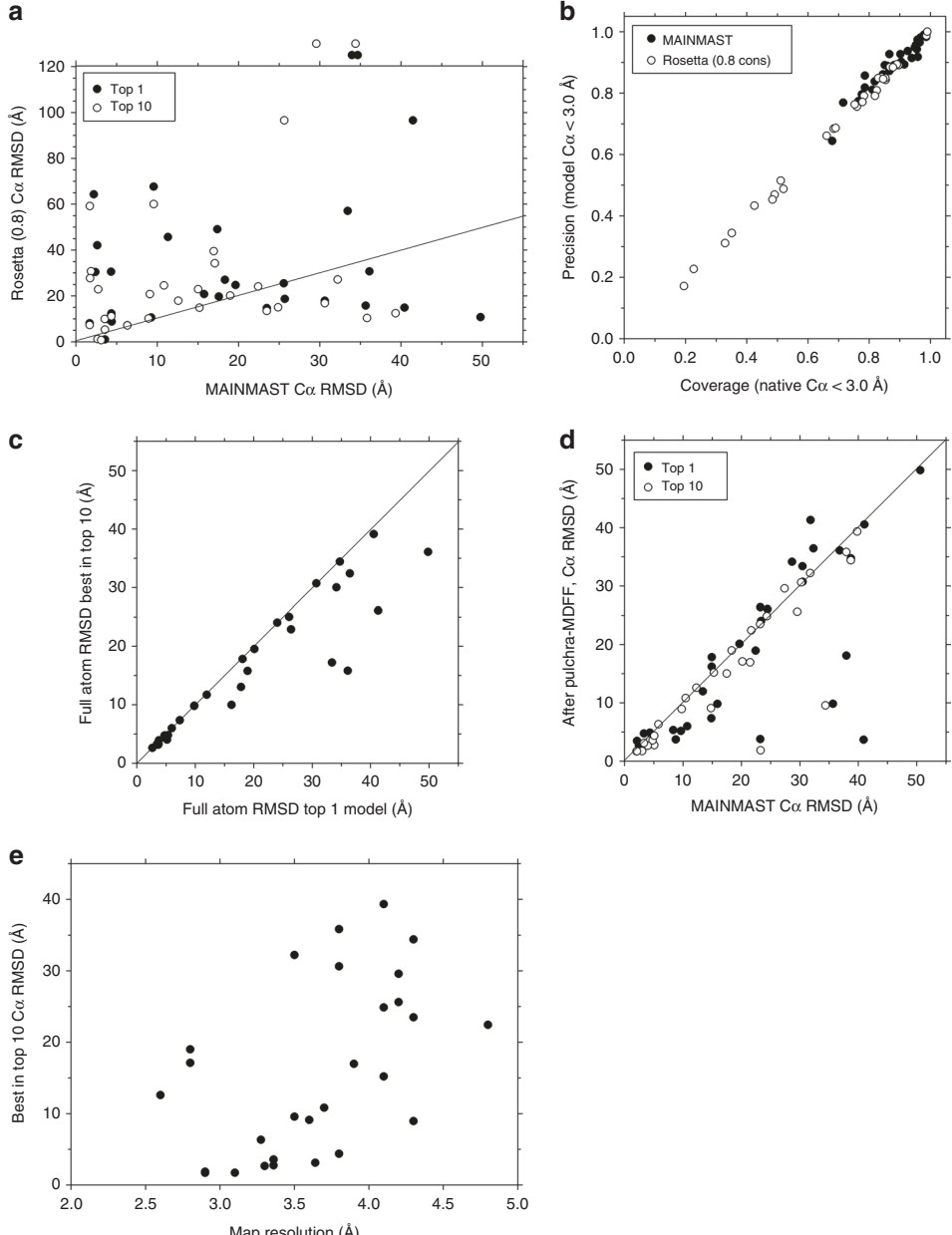

**Fig. 3** Modeling results of the 30 experimental EM maps by MAINMAST. **a** Cα RMSD of the top scoring (solid circles) and the best RMSD model among the top 10 scoring models (empty circles) by MAINMAST in comparison with the Rosetta top scoring models. The refinement by Pulchra and MDFF were applied to the models. For Rosetta, results using 0.8 for the consensus fraction was used, because it showed better results than the default setting (Supplementary Fig. 3). The points above the frame indicate that Rosetta could not model these proteins while MAINMAST made full models at the RMSD values. **b** Comparison between MAINMAST and Rosetta (with a 0.8 consensus setting) in terms of coverage and precision of models. Coverage (precision) is defined as the fraction of Cα atoms in the native structure (the model) which are closer than 3.0 Å to any Cα atoms in the model (the native structure). **c** Comparison between the top scoring (Top 1) model and the best RMSD model among the top 10 scoring model for each of the 30 EM maps. **d** Comparison of MAINMAST models before and after the refinement by Pulchra and MDFF. Models before the refinement were selected by the threading score while the scoring function of MDFF was used after the refinement. **e** RMSD of the models (the best among the top 10 socring models) by MAINMAST after refinement relative to the map resolution

top 1 model of F420-reducing hydrogenase α subunit (EMD-2513, the density map determined at a 3.36 Å resolution) and rotavirus VP6 capsid protein (EMD-6272, the map at 2.6 Å resolution), respectively, with a color code showing the degree of consensus among top 100 scoring models. The protein of EMD-2513 was modeled at an RMSD of 3.8 Å, which implies the modeling was reasonably successful overall; however, when examined closely, the topology of the N-terminus was incorrectly traced as shown in a magnified window on the left. The consensus

color code of this region is blue, indicating that this conformation was not supported by many alternative models. In contrast, high consensus regions (orange) are indeed well modeled as shown in the right window. Regarding the model built from EMD-6272 (Fig. 5b), the consensus color code indicates that the domain on the right is better modeled than the left domain. This is actually the case as the left domain in blue has incorrect connections in its β-sheet as shown in the magnified window, while the right domain with a higher consensus has nicely modeled helical

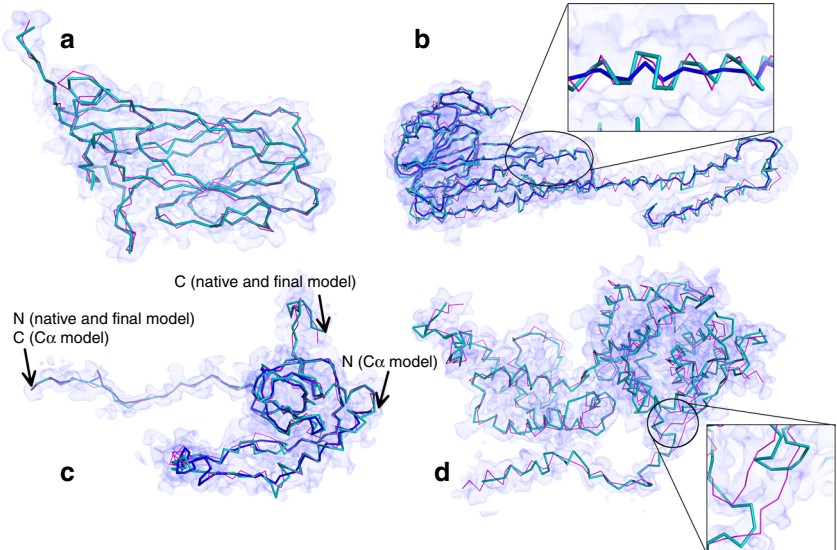

**Fig. 4** Examples of models generated by MAINMAST for experimental EM maps. The models were ranked first by the MDFF's score. Structures in turquoise are computed by MAINMAST and structures in magenta are the native structure. **a** capsid protein of porcine circovirus at 2.9 Å resolution (EMD-6555). MAINMAST model: 2.4 Å RMSD. Coverage and precision were both 0.88. Coverage and precision are defined as the fraction of Cα atoms in the native and the model, respectively, which are closer than 2.0 Å to any Cα atoms of the counterpart. Rosetta models have RMSDs of 31.6 and 30.4 Å with the default and a 0.8 consensus setting, respectively. Cov: 0.36, Prec: 0.36. **b** magnesium channel CorA at 3.8 Å resolution (EMD-6551). MAINMAST model (turquoise), RMSD: 4.4 Å, Cov: 0.91, Prec. 0.92. blue, the main-chain model prior to PULCHRA/MDFF refinement, RMSD: 10.7 Å, Cov: 0.81, Prec: 0.84. Rosetta models, 20.4/12.3 Å RMSD, Cov: 0.75/0.79, and Prec: 0.75/0.79 with the default/a 0.8 consensus setting. **c** eL6 protein from yeast 60S ribosomal subunit, at 2.9 Å resolution (EMD-6478). MAINMAST model, RMSD: 2.6 Å, Cov: 0.90, Prec: 0.90; blue, the main-chain model prior to PULCHRA/MDFF, RMSD:40.9 Å, Cov: 0.73, Prec: 0.74. This large RMSD is due to the failure of scoring a model with the correct sequence orientation by the threading score. However, a model with the correct sequence orientation was selected by MDFF after refinement. Rosetta models, 25.6/42.0 Å RMSD, Cov: 0.63/0.42, and Prec: 0.63/0.42 with the default/a 0.8 consensus settings. **d** helical Measles virus nucleocapsid protein at a 4.3 Å resolution (EMD-2867). MAINMAST model, RMSD: 9.3 Å, Cov: 0.68, Prec: 0.68; Rosetta models, RMSD: 21.3/10.6 Å, Cov: 0.68/0.72, Prec: 0.68/0.72 with the default/a 0.8 consensus setting

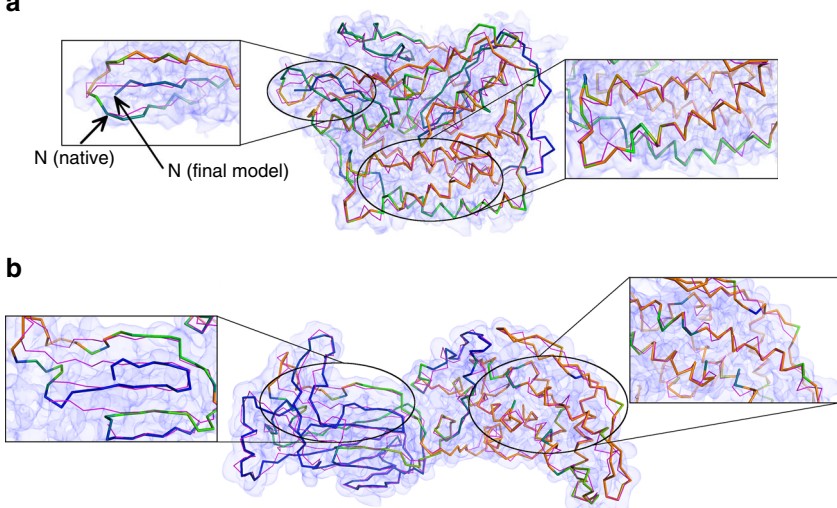

**Fig. 5** Models with confidence level in colors. **a** F420-reducing hydrogenase α subunit at 3.36 Å resolution (EMD-2513). The top-scoring MAINMAST full-atom model after the refinement had an RMSD of 3.8 Å, a coverage of 0.92, and a precision of 0.91 while the Cα model before the refinement was at an RMSD of 4.3 Å, a coverage of 0.88, and a precision of 0.88. The color code shows confidence of residue positions, which was computed by the degree of consensus among top 100 MDFF score models with blue to orange for low to high confidence regions. When only the residues that had consensus positions (within 3.5 Å) for over 50 models were considered (orange regions; 129 out of 385 residues), the RMSD was 2.1 Å. **b** rotavirus VP6 capsid protein at 2.6 Å resolution (EMD-6272). MAINMAST modeled it at 17.6 Å RMSD, Cov: 0.87, and prec: 0.86. Consensus residue positions over 50 models (orange regions) had an RMSD of 4.6 Å (180 out of 397 residues)

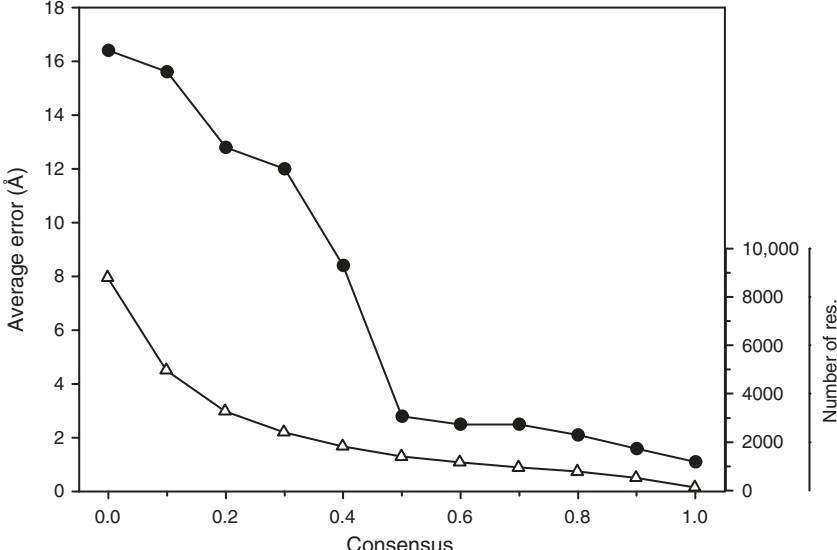

**Fig. 6** Average accuracy of residue positions relative to the degree of consensus among top 100 models for the 30 real EM maps. Cα positions of top 1 scoring model of 28 experimental EM maps were compared from those from top 2 -99 models, which were ranked by the MDFF score. Two maps, EMD-3073 and EMD-8116 were excluded because the top 1 protein models generated for these two maps were exceptionally bad (RMSD: 40.42 Å and 49.78 Å, respectively). Consensus on the x-axis shows the fraction of the models that have a residue within 3.5 Å. Black circles, the average error of each Cα positions (the bar on the left) of top 1 scoring model relative to the consensus fraction. Triangles, the total number of residues in the 30 models (the bar on the right) that have a certain consensus value. It is evident that the quality of regions with a consensus of 0.5 or higher are modeled well, on average within less than 3.0 Å

---

**Table 1 Comparison with RosettaES on the data set of 44 fragments**

| Criteria | Number of cases |
|---|---|
| Better or equal RMSD model | 16 |
| Better or RMSD difference ≤0.5 Å | 26 |
| Better or RMSD difference ≤1.0 Å | 32 |
| Better or RMSD difference ≤1.5 Å | 38 |
| Better or RMSD difference ≤2.0 Å | 40 |

The data set was taken from Supplementary Table 1 in the RosettaES paper[28]. These proteins have their EM maps of a 3-5 Å resolution and deposited structure models available in EMDB. The results of RosettaES and RosettaCM were taken from the columns of the best scoring results in the Supplementary Table 1 of the RosettaES paper. The number of cases among the 44 modeled fragments were counted where MAINMAST constructed a lower RMSD conformation to the native conformation than RosettaES or worse than RosettaES models but within a specified margin.

regions. Supplementary Fig. 4 shows that the error of Cα positions of these two models monotonically decreases as the degree of consensus increases. The correlation between the consensus degree and the accuracy is a general trend for MAINMAST models as shown in Fig. 6. It can be seen that the error decreases substantially, to around 3 Å on average, if a position is in consensus with 50 or more models.

**Building missing fragments**. We also compared MAINMAST with a recently published modeling protocol in the Rosetta package, RosettaES[28,] on the data set used in their paper. The data set consists of 44 segments of between 11–160 residues from nine proteins, which have EM maps of a 3-5 Å resolution and deposited tertiary structures (Supplementary Table 1 in the RosettaES paper[28]). The average backbone RMSD of the modeled fragments by RosettaES, RosettaCM[25], which was compared with RosettaES in their paper, and MAINMAST were 2.4, 10.3, and 2.68 Å, respectively. Table 1 shows summary of a direct comparison between RosettaES and MAINMAST. Out of 44 fragments in the data set, the model by MAINMAST had a lower or

identical RMSD with RosettaES models for 16 cases, and better or worse but within an RMSD margin of 0.5 or 1.0 Å to RosettaES models for 26 and 32 cases, respectively. Considering a recent study that investigated variability of structure models from EM maps[29], a difference within 1 Å is not meaningful for maps determined at a resolution of 3 Å. Thus, overall the performance of MAINMAST was comparable to RosettaES. The modeling results of individual fragments by MAINMAST are provided in Supplementary Table 4.

**Discussion**

We have developed MAINMAST, a fully automated de novo structure modeling method that constructs main-chain models from an EM density map. When benchmarked on 40 simulated EM maps and 30 experimental EM maps, the MAINMAST procedure showed better performance in comparison with other two existing methods, Pathwalking and Rosetta. In this work, Rosetta was run following the authors' tutorial and their paper[19] (Supplementary Note 1). It is noted that a completely fair comparison is not possible and different ways of running Rosetta may lead to better models. The purpose of the comparison between MAINMAST and the two existing methods is to characterize the strengths and weaknesses of MAINMAST.

We demonstrated that EM maps of around 4 Å resolution have sufficient information for MAINMAST to construct near native protein structure models and it can be accomplished without using any reference structures and or fragments. This is an advantage in that it generates a fully connected model, which is often a problem for methods that build models by assigning fragments. Another strength is that MAINMAST produces alternative models, which users can compare, examine, and choose from. Moreover, by computing consensus regions among top scoring models, MAINMAST provides a confidence level to structure regions in a model, which is shown to correlate well to the actual accuracy. On the other hand, MAINMAST has an intrinsic weakness that modeling can often have some difficulty at local low dense regions.

Structure modeling from an EM map requires different tools depending on the conditions, including map resolution and availability of homologous structures. Among such tools, de novo main-chain tracing methods, such as the one we presented, is one of the most fundamental and thus versatile. We expect the new method, MAINMAST, will serve as an indispensable tool for structure determination with EM.

## Methods

**The MAINMAST procedure.** MAINMAST consists of five steps (Fig. 1): (1) Identification of local dense points (LDPs) in an EM density map by the mean shifting algorithm; (2) Connection of LDPs to a MST; (3) Generating many tree structures by a tabu search; and (4) for each of the trees, aligning and evaluating the protein sequence in two directions by the threading score. (5) Finally, top scoring models by the threading score undergo refinement using PULCHRA and MDFF. Models are reranked by the score of MDFF.

**Identifying local dense points with mean shift.** The first step of the algorithm is to identify LDPs in an EM map, where protein atoms are more likely to exist. The mean shift algorithm[30], a non-parametric clustering algorithm originally developed for image processing, is used for this task. The assumption is that a density observed in a map is the sum of density functions that originate from atoms in the map. The primary assumption of the mean shift algorithm is that each point in the map represents a Gaussian density function and the local maxima of dense regions correspond to the chain positions of proteins. This process significantly reduces the number of points to consider in an EM map. For given grid points $x_i$ ($i = 1, ..., N$) of an EM map, initial seed points $y_j^{(0)}$ ($j = 1, ..., M$) are chosen whose density value is not less than a given threshold $\Phi_{thr}$. The seed points $y_j^{(0)}$ are iteratively updated, $y_j^{(t+1)} = f\left(y_j^{(t)}\right)$, as follows:

$$f(\mathbf{y}) = \frac{\sum_{n=1}^{N} k(\mathbf{y} - \mathbf{x_n})\Phi(\mathbf{x_n})\mathbf{x_n}}{\sum_{n'=1}^{N} k(\mathbf{y} - \mathbf{x_{n'}})\Phi(\mathbf{x_{n'}})}, \quad (1)$$

where $k(p)$ is a Gaussian kernel function and $\Phi(\mathbf{x})$ is a density value of the grid point $\mathbf{x}$. The $k(p)$ is defined as

$$k(p) = \exp\left(-1.5\left|\left|\frac{p^2}{\sigma}\right|\right|\right), \quad (2)$$

where the $\sigma$ is a bandwidth, which is set to 1.0.

After the seed point positions $y_j^{(t)}$ are updated, the density value of the points $\Theta(\mathbf{y})$ are computed as

$$\Theta(\mathbf{y}) = \frac{1}{N}\sum_{n=1}^{N} k(\mathbf{y} - \mathbf{x_n})\Phi(\mathbf{x_n}) \quad (3)$$

The density is further normalized with the minimum density value $\Theta_{min}$ and the maximum density value $\Theta_{max}$ of all the seed points

$$\theta(\mathbf{y}) = \frac{\Theta(\mathbf{y}) - \Theta_{min}}{\Theta_{max} - \Theta_{min}}. \quad (4)$$

Points are discarded if the density does not satisfy the threshold $\theta_{thr}$. Also, seed points that are closer than a threshold distance (0.5 Å) are clustered, and the highest density, $z_k$ in cluster $k$, is chosen as the representative of the cluster. This process is iterated until the positions of the selected representative points are converged. The representative points in the clusters are called LDPs. Finally in this step, LDPs $z_i$ and $z_j$ are labeled as adjacent if any pair of converged points, $y_n^{(t)}$ and $y_m^{(t)}$, in the two clusters where the LDPs belong to are originally adjacent at their initial seed point position, $y_n^{(0)}$ and $y_m^{(0)}$. This list of adjacent LDPs is used in the next step of connecting LDPs. Typically, the number of clusters is about 40% of the number of heavy atoms of the underlined protein in the map.

**Construction of MST.** The next step is to connect LDPs into a MST. MST is a graph structure that connects vertices with the minimal total weight of edges without forming cycles. Thus, a tree structure $T$ of MST minimizes

$$W(T) = \sum_{e \in T} w(e), \quad (5)$$

where $w(e)$ is a weight for an edge $e$. In MAINMAST, a weight of an edge is the Euclidean distance between connected LDPs by the edge. We adopt MST to connect LDPs with the minimum total distance because usually there are abundant LDPs along the main-chain of a protein. On top of the conventional algorithm to construct the MST, MAINMAST applies two heuristics: First, using the list of adjacent LDPs, two LDPs are connected only if they are in the list. Second, we also compute local MSTs for a local space defined by a sphere with a radius of $r_{local}$ centering at each LDP and

only edges in the local MSTs are considered in constructing the (global) MST. This is effective to improve the accuracy of local connection of the global MST.

**Refinement of tree structure.** The obtained MST is further refined because the longest path in the MST usually is not entirely correct with some wrong connections and disconnections. It turned out that the tree structure is further improved for finding the protein main-chain if branches of the tree, but not only the longest path, are also taken into account. Starting from the MST, a tree structure $T$ is evaluated with the following scoring function $S$:

$$S(T) = \sum_{n=1}^{N} \left(\sum_{e \in P_n} w'(e)\right)^2, \quad (6)$$

where $P_n$ is a $n$-th longest path in the given tree $T$, $e$ is an edge in $P_n$, and $w'(e)$ is a cost function of the edge $e$, which is defined as a product of the length and minimum value of $\Theta(\mathbf{y})$ on the edge $e$. $P_1$ is the longest path of the tree. The second longest path, $P_2$, is obtained by finding the longest path after removing all the edges in $P_1$, and so on. $N$ is the number of paths to be considered, which is set to 100.

Long branches are good candidates of partial paths to be included into the longest path during the iterative refinement.

Using the evaluation function in Eq. 6, the initial tree structure (i.e., MST) is refined in an iterative fashion using a tabu search[31]. A tabu search attempts to explore a large search space by keeping a list of moves that are visited recently and thus are forbidden (tabu list). Starting from a given tree (MST), in each iteration an existing edge $e_{delete}$ is deleted, which will split the tree into two parts, and then a different edge is added ($e_{add}$) to connect the two parts back to one tree. $e_{delete}$ is selected from possible moves that are not listed in the tabu list and should have a large weight, $w(e_{delete}) > d_{keep}$, to avoid a minor change in the resulting tree. At the same time, we restrict the move that gives a tree $T$ with the similar total length, $W(T)$ (Eq. 5), to the MST, i.e., $W(T) \leq 1.01 \times W(T_{MST})$.

This procedure is repeated for 30 times on an iteration and a tree that has the best score $S$ (Eq. 6) among the generated 30 trees is selected. Then, the newly generated tree overwrites the previous tree if the new one has a better score (Eq. 6). Thus, at the end of $N_{it}$ iterations, the tree that has the best score $S$ among generated at each iteration is kept.

Then, the chosen movement (i.e., the deleted edge and the added ege) and the movement that goes backwards from the updated tree to the original tree are added to the tabu list and the oldest edge pairs in the list are deleted. Since the size of the tabu list is set to 100 and two edges are added to the list at each iteration, edges are simply accumulated in the list for the first 50 iterations. The tabu search is efficient because the conformation search is performed by manipulating lists of edges.

**Threading target sequence on the longest path.** Using different combinations of parameters a pool of trees are generated, which are finally ranked by a scoring function that evaluates the fit of the amino acid sequence of the protein to a path in a tree (the threading score). The longest path of a tree is aligned with the expected density value of the amino acid sequence using the Smith–Waterman Dynamic Programming (DP) algorithm[32]. For a protein sequence, the density of each amino acid at the position $j$, $A_j$, is estimated by considering the average density of the amino acid in a set of simulated EM maps $D_{A_j}$ and a weight that depends on the predicted secondary structure $SS_j$, of the amino acid, $W_{SS_j}$:

$$A_j = W_{SS_j} D_{A_j} \quad (7)$$

The secondary structure of the protein sequence is predicted using SPIDER2[33]. On the other hand, for the longest path in the tree, the density value $V_j$ for a LDP $z_i$ is initially given by the sum of the density values of grid points that belong to the same cluster as $z_i$. which is then subject to smoothing by a Gaussian filter to obtain the final value as a weighted sum of neighboring LDPs:

$$V_k = \sum_{i=1}^{L} \left\{ v_k \frac{1}{\sigma_{path}} \exp\left(\frac{-||z_i - z_k||^2}{2\sigma_{path}}\right) \right\}, \quad (8)$$

where $L$ is the number of LDPs along the longest path, $\sigma_{path}$ is a standard deviation.

Thus, DP aligns the string of estimated densities of the amino acid sequence $A_j$ ($j = 1, .. N$; $N$ is the number of amino acids) and the string of densities of LDPs in the longest path $V_k$ ($k = 1, ... L$). The values of the densities are normalized by the Z-score computed relative to the densities of each string, which is used to define the similarity score $SCO_{i,j}$ for a pair of $A_j$ and $V_k$ to perform DP:

$$SCO_{i,j} = 1.00 - \left|Z(V_i) - Z(A_j)\right|, \quad (9)$$

where $Z(V_i)$ and $Z(A_j)$ are Z-score of $V_i$ and $A_j$, respectively. Using SCO, an alignment is computed with by the following rule to fill a DP matrix, $M$:

$$M(i,j) = \max \begin{cases} M(i, j-1) + gap \\ M(i-1, j-1) + w_{C\alpha}C\alpha\_dist(d) + SCO(i,j), \\ M(i-1, j) \end{cases} \quad (10)$$

where "gap" is a gap penalty for unassigned residues in the protein sequence, $w_{C\alpha}$ is a weight for a penalty score of C$\alpha$ distance (set to 0.9), C$\alpha$_dist:

$$C\alpha\_dist(d) = |d_{std} - d| \qquad (11)$$

where $d_{std}$ is the standard C$\alpha$-C$\alpha$ distance in protein chains and $d$ is the distance between the positions where residue $i-1$ and $i$ are aligned in the map. Note that in the third line in Eq. 10, a gap inserted to the amino acid sequence is not penalized because the number of LDPs is much larger than the number of amino acids in the protein (typically 1.6–3.5 times more than the protein length).

**Parameter combinations.** The whole procedure employs several parameters. Multiple C$\alpha$ models were constructed with different combinations of parameters. For the models of the 40 simulated maps we employed all combinations of the following parameters: The maximum number of the iterations ($N_{it}$): (10, 50, 100, 500); the threshold of the normalized density value ($\theta_{thr}$): 0.3; the constraint for the length ($d_{keep}$): (0.5, 1.0 Å); the threshold of the density value (thr): (9.0, 10.0, 11.0), the sphere radius of local MST ($r_{local}$): 5.0 Å. For each of the 24 ($= 4 \times 2 \times 3$) combinations of the parameters, ten trees were generated, from each of which a longest path was computed. For each of the ten paths, 112 ($= 7 \times 8 \times 2$ sequence directions) C$\alpha$ models were generated with combinations of $w_{bond}$: 0.9; $\sigma_{path}$: (0.8, 0.9, 1.0, 1.1, 1.2, 1.3, 1.4); $d_{std}$: (3.1, 3.2, 3.3, 3.4, 3.5, 3.6, 3.7, 3.8 Å), and two sequence directions. Thus in the overall process, $24 \times 10 \times 112 = 26{,}880$ C$\alpha$ models were generated. These C$\alpha$ models were ranked by their threading score. The top scoring models often include those which have overall similar conformations but with opposite sequence directions. In the results, up to top 10 best scoring models were reported in the result section.

For the data set of 30 experimental EM maps, we used combinations of the map density threshold ($\Phi_{thr}$): (the author recommended contour level $\times$ 0.5, 0.25) for Eq. 1; the density threshold ($\theta_{thr}$): (0.0, 0.1, 0.2, 0.3); the local map radius ($r_{local}$): (5.0, 7.5, 10.0 Å); and the edge weight threshold ($d_{keep}$): (0.5, 1.0, 1.5 Å). $N_{it}$ is set to 5000. The number of parameter combinations explored for the experimental maps were more than the simulated map cases, because each real EM map has a different resolution and electron density distribution. Similar to the simulated map cases, for each of the 72 ($= 2 \times 4 \times 3 \times 3$) parameter combinations ten trees, i.e., ten paths are generated, and the same 112 parameter combinations of $w_{bond}$, $\sigma_{path}$, $d_{std}$, and two sequence directions were used to generate in total of 80,640 C$\alpha$ models. The models were ranked by their threading score and top 500 models were selected.

These 500 models underwent full-atom structure building and structure refinement by PULCHRA[20] and MDFF[21] (g-scale: 0.5) and finally ranked by the scoring function by MDFF. The MDFF's score evaluates the molecular mechanics energy and the fit of a model to the EM map. Thus, it is highly possible that top-scoring models by MDFF are different from the threading scores' selection.

**Performance metrics.** Accuracy of a model relative to the native structure (the fitted structure for the EM map deposited to EMDB by the authors) was evaluated in three metrics. C$\alpha$ RMSD, RMSD computed between C$\alpha$ atom positions of the model relative to the corresponding C$\alpha$ atoms of the native structure. Thus, the amino acid sequence of the protein is taken into account. Coverage is defined as the fraction of C$\alpha$ atoms in the native structure that are within a certain distance cutoff (e.g., 2.0 Å) to any C$\alpha$ atoms in the model. Precision is computed from the opposite side; the fraction of C$\alpha$ atoms in the model that are within a certain distance cutoff (e.g., 2.0 Å) to any C$\alpha$ atoms in the native structure. Thus, coverage and precision consider the conformation of the model, but do not explicitly consider sequence information.

**Model building with Rosetta.** To construct a model for an EM map with the Rosetta package (ver. 3.6), first a fragment library was generated for a protein sequence that has 25 conformations per sequence position of 9 residue long using the Rosetta server (http://rosettaserver.bakerlab.org/). Fragments from homologous proteins were excluded. Next, following the tutorial of Rosetta tools, fragments were chosen and assembled in the EM map using the denovo_density program in the Rosetta package. Then, gaps in the structure that were not modeled by denovo_density were filled and refined using the RosettaCM protocol[25] following the paper by Wang et al[19]. Concrete commands used are provided in Supplementary Note 1. RosettaCM generates 1000 full-length models, from which the top 200 models in terms of the Rosetta energy were selected. Finally, the model that has the best agreement with the EM density (i.e., the lowest density score) was selected as the first model.

**Code availability.** The MAINMAST program is freely available for academic use through http://kiharalab.org/mainmast/index.html. Parameters used in the study are listed in the previous section, "Parameter combinations".

**Data availability.** The raw data of the structure models built by our method are provided in Supplementary Information, Supp. Table 1, 3, 4. The data that support the findings of this study are available from the corresponding author upon request.

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

## Acknowledgements

We acknowledge Lyman Monroe and Charles Christoffer for discussion and proof-reading the manuscript. This work was partly supported by the National Institutes of Health (R01GM123055, R01GM097528) and the National Science Foundation (IIS1319551, IOS1127027, DMS1614777).

## Author Contributions

G.T. and D.K. conceived the study. G.T. designed the MAINMAST procedure with D.K. and G.T.implemented the MAINMAST algorithm. Experiments were designed by G.T. and D.K. and were carried out by G.T. G.T and D.K. analyzed the results. The manuscript was drafted by G.T. D.K. administrated the project and wrote the manuscript.

## Additional information

**Competing interests:** The authors declare no competing interests.

