## [Peer Review File · Nature Communications]

Reviewers' comments:

Reviewer #1 (Remarks to the Author):

This paper describes a method called “MAINMAST” for automatically building a model based on a cryo-EM map in the resolution range of 3-4 Å. The core algorithm involves building a tree of connected points from the map, deriving a minimal spanning graph, interpreting that graph as a backbone trace, converting it to a full-atom model, and refining the model with molecular-dynamics-based approaches (MDFF). The method appears to work remarkably well, outperforming other new and powerful methods including Rosetta and Pathwalker. The paper is suitable for Nature Communications and the method is likely to be widely used.

The paper might be strengthened quite a bit in presentation if the authors could separate out the sequence comparison from the C-alpha positional comparisons. The reason this would be helpful is that two models can be almost identical in C-alpha positions but differ hugely once the connectivity and chain direction are taken into account.

Consider two models, one that is perfect, the other traced backwards. From the point of view of model-building such a pair of models can often be easily interconverted, so the two make a contribution of information that is nearly identical. A measure of model quality based on C-alpha positions will give nearly equal scores to the two models, as appropriate. From the point of view of the actual biological structure, of course the backwards model is totally incorrect. A score for matching the sequence of the model to the correct sequence can easily reflect this.

As it is, the scores used by the authors greatly penalize models that are traced backwards. Consequently presenting two sets of scores, one for C-alpha matching, the other for sequence matching, would make it much easier to see how close each model is to the known structure. This applies in particular to the scores of models built by the other methods.

Along the same lines, it would be helpful to discuss what is happening when, "the model was improved from 40.9 Å to 3.7 Å RMSD"? How much do individual atoms move? Is the connectivity changed?

Also along similar lines, in Fig 4, "2.9 Å (EMD-6478). MAINMAST model (green), 2.6 Å; blue, the main-chain model prior to PULCHRA/MDFF refinement, 40.9 Å. This large RMSD is due to the failure of mapping the protein sequence in a correct orientation on the model main-chain

path.” How does pulchra/mdff change the orientation of the model main_chain path? Pulchra doesn’t know which way the main chain goes. Do you score first with a random direction, try pulchra each way, score both, and pick the best scoring one? (That would be fine but it would be nice to know what is happening).

Reviewer #2 (Remarks to the Author):

This is an interesting manuscript that is timely given the rapid increase in the number of higher resolution cryo-EM reconstructions in the last 2 years. The authors correctly argue that reliable automated map interpretation tools are needed to support the analysis of these datasets. The description of the MAINMAST method, and analyses of its performance on a set of simulated and real maps is of great interest and worthy of publication. The use of graph analytical methods is novel in this context and clearly has some potential, as does the use of the threading score to help assign sequence. These methods could also be effectively combined with other methods to make even better protocols. However, a significant fraction of the manuscript is devoted to comparisons with other model building methods. Unfortunately, in its current form this material significantly reduces the value of the manuscript. Here are points the authors should consider:

- The comparison of the method to results obtained from other software is fraught with problems:

1. How were the models created from Pathwalker obtained? There are no methods describing how Pathwalker was run. This needs to be clarified.
2. Why compare to Pathwalker results from 2012 and 2016? It seems clear that the 2016 results supersede the 2012 results - this seems evident from Figure 2A/2B. A fairer comparison would be solely with the 2016 results.
3. The methods section that describes running Rosetta mentions Rosetta version 3.4. This is a version from 2012, which significantly predates the cryo-EM map interpretation methods developed by DiMaio and others. In addition, the description of how Rosetta was run (using RosettaCM alone) is at odds with current practice. There are specific de novo density interpretation tools that researchers would use to create models. I think it possible that the Rosetta results are not reflective of the current state of technology because an out-of-date procedure (and software version) was used. For a reasonable comparison, the Rosetta results need to be performed with something closer to what is available to the research community.
4. While there is some value to comparing methods implemented in programs, I’m sure the authors appreciate that this is a non-trivial task. One of the challenges is keeping the tests current

with what the community has ready access to. If the authors wish to make use of comparisons to other packages it seems reasonable that they perform tests with versions that the research community are able to use - which is challenging given the rapid change in the cryo-EM technology. An alternative would be to minimize comparisons and instead focus on the absolute performance of the method as it is described.

- It isn't completely clear how the various performance metrics were calculated. In particular, does the C-alpha metric take into account the sequence, or is it just the placement of any C-alpha atom? The authors could greatly help the reader by providing a section in the methods that describes the different metrics and how they were calculated.

- The stated resolutions of the experimental maps analyzed is incorrect in the Abstract (although correct in the body of the manuscript). The resolution range tested is closer to ~3-5, with 6 structures better than 3A, 15 structures between 3 and 4A, and only 9 between 4A and 5A. I suggest that the authors reword the abstract to make this clear, as it greatly influences the reader's understanding of what has been achieved.

Responses to Comments by Reviewer #1:

This paper describes a method called “MAINMAST” for automatically building a model based on a cryo-EM map in the resolution range of 3-4 Å. The core algorithm involves building a tree of connected points from the map, deriving a minimal spanning graph, interpreting that graph as a backbone trace, converting it to a full-atom model, and refining the model with molecular-dynamics-based approaches (MDFE). The method appears to work remarkably well, outperforming other new and powerful methods including Rosetta and Pathwalker. The paper is suitable for Nature Communications and the method is likely to be widely used.

Thank you.

The paper might be strengthened quite a bit in presentation if the authors could separate out the sequence comparison from the C-alpha positional comparisons. The reason this would be helpful is that two models can be almost identical in C-alpha positions but differ hugely once the connectivity and chain direction are taken into account.

Thank you very much for the comment. We totally agree with the reviewer’s opinion.

Indeed, results of the sequence-free C-alpha positional accuracy are shown. In the results shown for the 40 simulated maps (Figure 2), Figure 2b and 2d are for coverage, which is defined as the fraction of C-alpha atoms in the protein structure that are covered by a model within a cutoff distance. For Figure 2b we used the CLICK server, because the Pathwalking results were taken

from their papers where the authors used CLICK. The explanations of the coverage in the text was somewhat vague, and thus we revised them. The revised explanation is in page 8, 9, and 10.

For the results of the experimental maps (Figure 3), Figure 3b is the sequence-alignment-free Calpha position comparison. Coverage on the X-axis is the fraction of Calpha atoms in the native structure that are closer than 3 Angstroms to any residues in the compared model while the y-axis, precision, computes the same type of value from the model side, i.e. the fraction of Calpha atoms in the model that are closer than 3 Angstroms to any residues in the native structure. Again we revised the description of coverage and precision in page 13 and 15. Supplementary Table 1 and 3 also provide coverage data for the generated models.

In addition, we added a new section in Method with a subtitle of “Performance metrics” (page 34) to clarify the 3 metrics, RMSD, coverage, and precision.

Consider two models, one that is perfect, the other traced backwards. From the point of view of model-building such a pair of models can often be easily interconverted, so the two make a contribution of information that is nearly identical. A measure of model quality based on C-alpha positions will give nearly equal scores to the two models, as appropriate. From the point of view of the actual biological structure, of course the backwards model is totally incorrect. A score for matching the sequence of the model to the correct sequence can easily reflect this.

In Figure 4, Figure 4c is exactly the case that a model chain conformation is almost correct but the sequence was mapped on the opposite direction. In the case of Figure 4C, RMSD of the model was very large, 40.9 Angstrom, while the coverage was 0.73, not as bad as it may seem from the RMSD value. We mentioned it in the text in page 17. We have also provided the coverage and precision values for all the models in Figure 4, not only for Fig. 4c.

As it is, the scores used by the authors greatly penalize models that are traced backwards. Consequently presenting two sets of scores, one for C-alpha matching, the other for sequence matching, would make it much easier to see how close each model is to the known structure. This applies in particular to the scores of models built by the other methods.

As answered in the comments above, the coverage of models are provided in Figure 2 (for simulated maps) and Figure 3 (experimentally determined maps) as well as Supplementary Table 1 and 3. Figure 4 caption also now has coverage values.

Along the same lines, it would be helpful to discuss what is happening when, " the model was improved from 40.9 Å to 3.7 Å RMSD"? How much do individual atoms move? Is the connectivity changed?

Thank you for pointing it out, it was not well described. This improvement happened because different models were selected by different scoring functions before and after the refinement. Since models output by MAINMAST only contain Calpha atoms, Calpha models were ranked by the threading score that matches the sequence information to local density along the main-chain. But after the all-atom building by PULCHRA and refinement by MDFF, we used the scoring function in MDFF. We added the explanation in the caption of Figure 3 and in the main text that

discussed Fig. 3d (page 13, bottom). Also, we added the explanation in page 5-6, where the flowchart of the overall procedure was mentioned. It reads “The top 500 Models selected by the threading score are subject to full-atom reconstruction and refinement using PULCHRA²⁰. Finally, the full-atom models are refined using Molecular Dynamics Flexible Fitting (MDFF)²¹, a molecular dynamics-based method, and selected according to the scoring function implemented in MDFF.”

Also along similar lines, in Fig 4, "2.9 Å (EMD-6478). MAINMAST model (green), 2.6 Å; blue, the main-chain model prior to PULCHRA/MDFF refinement, 40.9 Å. This large RMSD is due to the failure of mapping the protein sequence in a correct orientation on the model main-chain path." How does pulchra/mdff change the orientation of the model main_chain path? Pulchra doesn't know which way the main chain goes. Do you score first with a random direction, try pulchra each way, score both, and pick the best scoring one? (That would be fine but it would be nice to know what is happening).

The protein sequence is mapped in two orientations on the longest path of a tree structure and each of them are evaluated by the threading score before Calpha models were passed to Pulchra and MDFF refinement. Many trees were generated by different combinations of parameters, and for each tree two orientations were tested. Then, finally top 500 Calpha models by the threading score were passed to the refinement. Thus, the 500 Calpha models will include models of different sequence mapping directions. As you wrote, Pulchra does not know the sequence direction for a protein main-chain path. Correcting the orientation happened because the MDFF score selected a model with a correct orientation as its top choice.

This is described in the Method section (page 33-34) but we modified the explanation to make it clearer.

Also, in page 16, which had the particular sentence of “This large RMSD ..” was revised to “This large RMSD is due to the failure of scoring a model with the correct sequence orientation by the threading score. However, a model with the correct sequence orientation was selected by MDFF after refinement.” In addition, the description of this Figure (Fig. 4c) in the main text was also revised (page 17).

Responses to Comments by Reviewer #2:

This is an interesting manuscript that is timely given the rapid increase in the number of higher resolution cryo-EM reconstructions in the last 2 years. The authors correctly argue that reliable automated map interpretation tools are needed to support the analysis of these datasets. The description of the MAINMAST method, and analyses of its performance on a set of simulated and real maps is of great interest and worthy of publication. The use of graph analytical methods is novel in this context and clearly has some potential, as does the use of the threading score to help assign sequence. These methods could also be effectively combined with other methods to make even better protocols. However, a significant fraction of the manuscript is devoted to comparisons with other model building methods. Unfortunately, in its current form this material significantly reduces the value of the manuscript. Here are points the authors should consider:

- The comparison of the method to results obtained from other software is fraught with problems:

1. How were the models created from Pathwalker obtained? There are no methods describing how Pathwalker was run. This needs to be clarified.

Data of Pathwalking were taken from their publications: Version 2016 was taken from

•Chen, M., Baldwin, P. R., Ludtke, S. J. & Baker, M. L. De Novo modeling in cryo-EM density maps with Pathwalking. *J Struct Biol* **196**, 289-298, (2016).

And data for version 2012 were taken from

•Baker, M. R., Rees, I., Ludtke, S. J., Chiu, W. & Baker, M. L. Constructing and validating initial Calpha models from subnanometer resolution density maps with pathwalking. *Structure* **20**, 450-463, (2012).

It was stated in the caption of Figure 2, “For the Pathwalking algorithm, data are taken from the publication in 2016.” but we now also placed this statement in the main text at page 8. We used the published data for Pathwalking because the program version 2016 is not made available.

2. Why compare to Pathwalker results from 2012 and 2016? It seems clear that the 2016 results supersede the 2012 results - this seems evident from Figure 2A/2B. A fairer comparison would be solely with the 2016 results.

We removed data of Pathwalking ver. 2012 data from Fig. 2 as requested. It is moved to the Supplemental Figure 1.

3. The methods section that describes running Rosetta mentions Rosetta version 3.4. This is a version from 2012, which significantly predates the cryo-EM map interpretation methods developed by DiMaio and others. In addition, the description of how Rosetta was run (using RosettaCM alone) is at odds with current practice. There are specific de novo density interpretation tools that researchers would use to create models. I think it possible that the Rosetta results are not reflective of the current state of technology because an out-of-date procedure (and software version) was used. For a reasonable comparison, the Rosetta results need to be performed with something closer to what is available to the research community.

The version of Rosetta we used was ver. 3.6. It was a mistake that we put ver. 3.4, and now it is fixed (page 36).

We ran Rosetta as indicated in the Rosetta tutorial file, which was provided by the Dimaio lab: <http://dimaiolab.ipd.uw.edu/software/>. The tutorial provided is basically consistent with the way Rosetta was run in the paper “De novo protein structure determination from near-atomic-resolution cryo-EM maps”, Wang et al., *Nature Methods*, 12: 335-338, (2015), where the structural modeling of Rosetta for EM maps was described. To concretely clarify how Rosetta was run, we provided a list of commands and parameters used in the run in the Supplemental Material (the subsection named “Running Rosetta”; the last pages of the Supplemental Material).

Although we followed the tutorial and the Rosetta paper to run the Rosetta program, I agree with the reviewer that it is possible that software may be run in a different way that could lead to a better results. A perfectly fair comparison is not possible, but in our experience as structure modeling/prediction developers, comparison is useful to clarify strengths and weaknesses of the proposed new method, acknowledging that the perfectly fair comparison is not possible.

One unique advantage of MAINMAST is that it does not need any manual parameter tunings and human intervention in the process of structure modeling. Thus, although some other ways may exist, comparing Rosetta's results by following authors' tutorial maybe not too unreasonable comparison.

We added new discussion in the Discussion (page 23) to clarify that how we run Rosetta and that the Rosetta could run in a different way which could achieve better models. It reads:

In this work, Rosetta was run following the authors' tutorial the paper¹⁹ (see the setting used in Supplemental Materials). It is noted that a completely fair comparison is not possible and different way of running Rosetta may lead to better models. The purpose of the comparison between MAINMAST and the two existing methods is to characterize the strengths and weaknesses of MAINMAST.

Comparison with Rosetta is not for us to claim the superiority of our method but as a part of characterizing our method and show that our method is another method that users can try to use as a complementary method of the field. The comparison with existing methods is also partly requested by the editorial board of this journal.

4. While there is some value to comparing methods implemented in programs, I'm sure the authors appreciate that this is a non-trivial task. One of the challenges is keeping the tests current with what the community has ready access to. If the authors wish to make use of comparisons to other packages it seems reasonable that they perform tests with versions that the research community are able to use - which is challenging given the rapid change in the cryo-EM technology. An alternative would be to minimize comparisons and instead focus on the absolute performance of the method as it is described.

We totally agree that a perfectly fair comparison of methods is very difficult, if not impossible, because methods are modified day by day as pointed out, and also because parameters and setting of each method were optimized on a certain EM map dataset, which maybe overlapping with the dataset used for evaluation. As we answered to the pervious comment by the reviewer, while we know the limitation of the comparison, comparison would be informative for characterizing features of the new method. We acknowledged the difficulty of the fair comparison, and specified in the details how Rosetta was run.

As the reviewer says, we are more interested in showing analysis our own absolute performance than comparison (but at the same time we feel we need to show minimum sufficient level of comparisons because there are existing methods). Indeed, in figures and tables in the manuscript, more figures were spent for reporting our method's performance than for comparison: Fig. 1, 4, 5, are for analysis of our own methods, while Table 1 is for a comparison, 2 out of 4 of panels in Figure 2 and 3 out of 5 panels in Figure 3 are for analyzing our method.

- It isn't completely clear how the various performance metrics were calculated. In particular, does the C-alpha metric take into account the sequence, or is it just the placement of any C-alpha atom? The authors could greatly help the reader by providing a section in the methods that describes the different metrics and how they were calculated.

I agree that some of the descriptions of the metrics in the text were not very clear. We have fixed those. Also, as suggested, we added a new section in Method titled "Performance metrics". In that new section, we explained RMSD, coverage, and precision used to evaluate model quality. In short, the RMSD considers the sequence while the coverage and precision are sequence-free metrics. The latter two metrics are able to detect models which have as Reviewer 1 pointed out.

- The stated resolutions of the experimental maps analyzed is incorrect in the Abstract (although correct in the body of the manuscript). The resolution range tested is closer to ~3-5, with 6 structures better than 3Å, 15 structures between 3 and 4Å, and only 9 between 4Å and 5Å. I suggest that the authors reword the abstract to make this clear, as it greatly influences the reader's understanding of what has been achieved.

Thank you for pointing it out. We corrected it to "at 2.6 to 4.8 Å resolution".

Reviewers' Comments:

Reviewer #1 (Remarks to the Author):

The authors have suitably addressed all the comments of myself and the other reviewer. The paper is quite appropriate for publication.

Reviewer #2 (Remarks to the Author):

The authors have made good progress in addressing many of the issues raised by the reviewers. However, I am still unconvinced with the Rosetta comparison. It is my understanding that the relevant tool in Rosetta is denovo_density (described online by the authors of that tool; https://faculty.washington.edu/dimaio/files/rosetta_density_tutorial_may17_4.pdf):

“In this scenario, we introduce a tool, denovo_density, aimed at automatically building backbone and placing sequence in 3-4.5 Å cryoEM density maps. This tool is primarily intended for cases where a model is to be built with no known structural homologues. It is relatively expensive computationally, and consists of four basic steps:

- Search for local backbone "fragments" in the density map
- Score the "compatibility" of sets of placed fragments
- Monte Carlo sampling for the "maximally compatible" fragment set
- Consensus assignment from the best-scoring Monte Carlo trajectories”

This would appear to be the appropriate comparison, rather than RosettaCM. It may be a shortcoming in the description of the methods in the updated manuscript, but either way, the situation needs to be corrected and clarified.

Responses to Reviewers:

Response to Comment by Reviewer #1:

The authors have suitably addressed all the comments of myself and the other reviewer. The paper is quite appropriate for publication.

Thank you very much.

Response to Comment by Reviewer #2:

*The authors have made good progress in addressing many of the issues raised by the reviewers. However, I am still unconvinced with the Rosetta comparison. It is my understanding that the relevant tool in Rosetta is *denovo_density* (described online by the authors of that tool; https://faculty.washington.edu/dimaio/files/rosetta_density_tutorial_may17_4.pdf):*

*“In this scenario, we introduce a tool, *denovo_density*, aimed at automatically building backbone and placing sequence in 3-4.5 Å cryoEM density maps. This tool is primarily intended for cases where a model is to be built with no known structural homologues. It is relatively expensive computationally, and consists of four basic steps:*

- Search for local backbone "fragments" in the density map*
- Score the "compatibility" of sets of placed fragments*

- Monte Carlo sampling for the "maximally compatible" fragment set
- Consensus assignment from the best-scoring Monte Carlo trajectories"

This would appear to be the appropriate comparison, rather than RosettaCM. It may be a shortcoming in the description of the methods in the updated manuscript, but either way, the situation needs to be corrected and clarified.

Actually, we have done exactly what was indicated above. We followed the tutorial, and in the previous revision of the supplemental data, we added a list of commands in Rosetta we used from page 13 to 15. The steps were

1. Local fragment search
2. Placed fragment scoring
3. Monte Carlo Fragment assembly
4. Consensus assignment
5. Running RosettaCM

Here for 1-4, as indicated the actual command used, we ran "\$ROSETTA3/source/bin/denovo_density.linuxgccrelease". This is the Linux version of denovo_density compiled by gcc (C compiler). In this revision, to clarify that 1-4 uses denovo_density, I added "using denovo_density" to the title of the steps. We also added one-line description at each step.

The 5th step, running RosettaCM, is to fill main-chain gaps that were not modelled by fragment assembly by denovo_density. This is how Rosetta was used in the following paper we compared with in figures and supplemental tables: "Wang, R. Y. R., Kudryashev, M., Li, X., Egelman, E. H., Basler, M., Cheng, Y., Baker, D., & DiMaio, F. (2015). De novo protein structure determination from near-atomic-resolution cryo-EM maps. *Nature Methods*, 12(4), 335-338."

In the main text, we now have also explicitly mentioned denovo_density to clarify it and avoid confusion.